# CYCLON and NPM1 Cooperate within an Oncogenic Network Predictive of R-CHOP Response in DLBCL

**DOI:** 10.3390/cancers13235900

**Published:** 2021-11-24

**Authors:** Antonin Bouroumeau, Lucile Bussot, Sieme Hamaidia, Andrea Garcìa-Sandoval, Anna Bergan-Dahl, Patricia Betton-Fraisse, Samuel Duley, Cyril Fournier, Romain Aucagne, Annie Adrait, Yohann Couté, Anne McLeer, Edwige Col, Laurence David-Boudet, Tatiana Raskovalova, Marie-Christine Jacob, Claire Vettier, Simon Chevalier, Sylvain Carras, Christine Lefebvre, Caroline Algrin, Rémy Gressin, Mary B. Callanan, Hervé Sartelet, Thierry Bonnefoix, Anouk Emadali

**Affiliations:** 1Department of Pathology, Grenoble-Alpes University Hospital, 38000 Grenoble, France; Antonin.Bouroumeau@hcuge.ch (A.B.); AMcLeer@chu-grenoble.fr (A.M.); ecol@chu-grenoble.fr (E.C.); LDavid-boudet@chu-grenoble.fr (L.D.-B.); H.SARTELET@chru-nancy.fr (H.S.); 2Translational Epigenetics, Institute for Advanced Biosciences, INSERM U1209, CNRS UMR 5309, Grenoble-Alpes University, 38000 Grenoble, France; lbussot@chu-grenoble.fr (L.B.); sieme.hamaidia@univ-grenoble-alpes.fr (S.H.); a.c.garcia.sandoval@gmail.com (A.G.-S.); anna.bd@hotmail.com (A.B.-D.); patricia.betton@univ-grenoble-alpes.fr (P.B.-F.); samuel.duley@univ-grenoble-alpes.fr (S.D.); SCarras@chu-grenoble.fr (S.C.); CLefebvre@chu-grenoble.fr (C.L.); RGressin@chu-grenoble.fr (R.G.); thierry.bonnefoix@univ-grenoble-alpes.fr (T.B.); 3Division of Clinical Pathology, Diagnostic Department, Hôpitaux Universitaires Genève, 1205 Geneva, Switzerland; 4Department of Clinical Hematology, Grenoble-Alpes University Hospital, 38000 Grenoble, France; 5Pole Recherche, Grenoble-Alpes University Hospital, 38000 Grenoble, France; 6Unit for Innovation in Genetics and Epigenetics in Oncology and Dijon University Hospital, University of Bourgogne, INSERM 1231, 21000 Dijon, France; cyril.fournier@chu-dijon.fr (C.F.); romain.aucagne@u-bourgogne.fr (R.A.); mary.callanan@chu-dijon.fr (M.B.C.); 7Institut National de la Santé et de la Recherche Médicale (INSERM), University Grenoble Alpes, CEA, UMR BioSanté U1292, CNRS, CEA, FR2048, 38000 Grenoble, France; annie.adrait@cea.fr (A.A.); yohann.coute@cea.fr (Y.C.); 8Hematology, Oncogenetics and Immunology Unit, Grenoble-Alpes University Hospital, 38000 Grenoble, France; TRaskovalova@chu-grenoble.fr (T.R.); MCJacob@chu-grenoble.fr (M.-C.J.); CVettier@chu-grenoble.fr (C.V.); schevalier@chu-grenoble.fr (S.C.); 9Daniel Hollard Institute, 38000 Grenoble, France; caroline.algrin@avec.fr; 10Department of Pathology, Nancy Regional University Hospital, 54000 Nancy, France

**Keywords:** NPM1, CYCLON, nucleolus, R-CHOP, prognosis, R-IPI

## Abstract

**Simple Summary:**

CYCLON is a nuclear protein, which has been associated with disease progression and treatment resistance in DLBCL, the most common form of aggressive B-cell lymphoma, but also represents a predictive factor of refractory disease and relapse for immuno-chemotherapy-treated DLBCL patients. The molecular mechanisms related to this unstructured protein remain largely uncharacterized. Here, we performed a mass-spectrometry-based identification of the CYCLON protein interactome that suggested it could exert nucleolar functions related to cell proliferation. Among the CYCLON oncogenic network, we performed an immunohistochemical evaluation of the multi-functional nucleolar protein NPM1 in a DLBCL cohort and showed that CYCLON/NPM1 concomitant expression delineates a poor prognosis subgroup of patients. Multivariate survival analyses demonstrated that specific sub-cellular localizations of CYCLON and NPM1 represent independent novel predictors specifically associated with refractory DLBCL.

**Abstract:**

R-CHOP immuno-chemotherapy significantly improved clinical management of diffuse large B-cell lymphoma (DLBCL). However, 30–40% of DLBCL patients still present a refractory disease or relapse. Most of the prognostic markers identified to date fail to accurately stratify high-risk DLBCL patients. We have previously shown that the nuclear protein CYCLON is associated with DLBCL disease progression and resistance to anti-CD20 immunotherapy in preclinical models. We also recently reported that it also represents a potent predictor of refractory disease and relapse in a retrospective DLBCL cohort. However, only sparse data are available to predict the potential biological role of CYCLON and how it might exert its adverse effects on lymphoma cells. Here, we characterized the protein interaction network of CYCLON, connecting this protein to the nucleolus, RNA processing, MYC signaling and cell cycle progression. Among this network, NPM1, a nucleolar multi-functional protein frequently deregulated in cancer, emerged as another potential target related to treatment resistance in DLBCL. Immunohistochemistry evaluation of CYCLON and NPM1 revealed that their co-expression is strongly related to inferior prognosis in DLBCL. More specifically, alternative sub-cellular localizations of the proteins (extra-nucleolar CYCLON and pan-cellular NPM1) represent independent predictive factors specifically associated to R-CHOP refractory DLBCL patients, which could allow them to be orientated towards risk-adapted or novel targeted therapies.

## 1. Introduction

Diffuse large B-cell lymphoma (DLBCL) is a hematological malignancy deriving from the transformation of a mature B lymphocyte. It represents the most common subtype (40%) of non-Hodgkin’s B lymphoma (B-NHL) [1]. Currently, the standard first-line DLBCL treatment is based on an immuno-chemotherapy combining an anti-CD20 monoclonal antibody (Rituximab) with a CHOP-type poly-chemotherapy (including cyclophosphamide, doxorubicin, vincristine and prednisolone). Although this treatment allows a high response rate with approximately 60–70% of patients achieving a complete response [2], the remaining 30–40% of patients present a refractory disease or relapse, usually within two years. This outcome discrepancy can be explained by the high degree of clinical and biological heterogeneity of DLBCL [3].

This heterogeneity is related to a wide variety of genomic alterations and phenotypic features that include: (i) the cell-of-origin (COO) gene expression signatures defining “germinal center B cells” (GCB) or “activated B cells” (ABC) DLBCL subtypes, the latter being associated with an inferior prognosis [4,5] but not correlated with R-CHOP response, (ii) gene translocations involving *MYC* and *BCL2/BCL6* and defining the refractory subtype “double/triple hit” (DH/TH), now defined as a distinct category in the updated 2016 WHO Classification of Hematopoietic and Lymphoid Tissues termed high-grade B-cell lymphoma (HGBL) and iii) recently identified recurrent mutational events defining genetic subtypes that could have therapeutic implications [6,7,8]. Despite this extensive molecular characterization, the only robust DLBCL prognostic factor remains the revised version of the International Prognostic Index (R-IPI), combining five clinical and biological readouts [9]. However, R-IPI remains insufficient to recapitulate disease complexity and fails to accurately discriminate refractory/relapse (R/R) DLBCL cases.

In this context, we have previously shown that CYCLON, a novel nuclear phosphoprotein was acting as an autonomous tumor growth driver and anti-CD20 treatment resistance factor in DLBCL in vitro and in a xenograft mouse model [10]. We have also recently demonstrated that CYCLON sub-nuclear localization represents an independent robust predictive factor in a retrospective cohort of R-CHOP treated DLBCL [11]. CYCLON is a coiled-coil protein, mainly expressed in male germ cells [10,12]. However, it has also been previously shown to be involved in various contexts in immune cells: response to cytokines (IL-3) induction in mouse B cells [12], T cells apoptosis by upregulation of Fas [13], and in B cells that undergo proliferation during germinal center selection [14]. Overall, CYCLON appears to play a role in high proliferation contexts, which is consistent with its identification as a MYC downstream target [10,14]. CYCLON does not carry any specific domain that could help to define its molecular function. Transcriptomic analysis of CYCLON knock-down lymphoma cells suggest that CYCLON drives the expression of a male germ cell program in lymphoma cells related to tumor aggressiveness [10]. However, the mechanisms by which CYCLON could control gene expression and exert these adverse effects in DLBCL remain unknown. To get further insights into the actual functions of this new B-cell lymphoma marker and identify members of its oncogenic network that could represent novel predictive factors of DLBCL clinical outcome, we characterized the CYCLON protein interactome and identified NPM1 as a joint marker that further improves CYCLON-based stratification of high-risk DLBCL patients.

## 2. Materials and Methods

### 2.1. Cell Lines

The B593 DLBCL cell line was derived in our laboratory [15]. Cells were cultured in RPMI medium supplemented with 2 mM GlutaMAX^®^, 1 mM sodium pyruvate, 25 mM HEPES, non-essential amino acids, 100 mg/mL penicillin/streptomycin solution and 20% fetal calf serum. Cells were grown at 37 °C, 5% CO2. Cell culture reagents were purchased from Thermo Fisher Scientific (Illkirch, France). Cells were kept at a density between 0.3 and 1.5 million cells/mL (passage every 2–3 days). Cell viability was estimated with ethidium bromide and acridine orange double staining. B593 GFP-CYCLON cells were produced via transduction of B593 with a lentiviral vector designed to express CYCLON with an *N*-terminus GFP-tag. B593 GFP-CYCLON cells were derived from the polyclonal cell population by limiting dilution and pooling the 3 clones showing the lower GFP expression to generate an oligoclonal cell line in which the CYCLON-GFP expression level is close to the endogenous level of the protein.

### 2.2. Nuclei Isolation

Nuclei were isolated via hypotonic cell lysis from 100 million cells using 2 mL of nucleic buffer-NB (60 mM KCl, 15 mM NaCl, 5 mM MgCl2, 0.1 mM EGTA pH8, 15 mM Tris pH 8, 0.3 M sucrose, 1 mM DTT, 1× complete protease inhibitor cocktail, 1× PIC3 phosphatase inhibitor cocktail (Sigma Aldrich, Saint-Quentin-Fallavier, France), 1 µg/mL TSA, 0.2% NP40). After 10 min incubation on ice, detergent was diluted with 8 mL of NB without NP40 before centrifugation (2000 rpm, 10 min, 4 °C). Nuclei were then washed in HDG150 buffer (20 mM HEPES pH7.4, 150 mM KCl, 1 mM DTT, 10% glycerol, 1× complete protease inhibitor cocktail). Nuclei integrity was verified using trypan blue staining under an inverted light microscope (Eclipse TS2 FL Statif Inverse Fluo, Nikon, Champigny sur Marne, France).

### 2.3. Solubilization of Nucleic-Acid Bound Complexes Using Micrococcal Nuclease (MNase)

Nuclei were quickly sonicated using a Vibracell ultrasonic processor (20% amplitude for 5 s with 2 s pulse, 2 s pause) before DNA quantification using NanoDrop (Ozyme, Saint-Cyr-l’École, France). DNA digestion was performed via addition of 1 U MNase and 1 mM CaCl_2_ per 10 µg chromatin before incubation at 37 °C for 15 min with agitation every 2 min (until most of the chromatin was digested as mononucleosomes, visualized using migration on a 2% agarose gel electrophoresis). Digestion was then stopped by addition of 10 mM EDTA. Digested fraction was incubated at 4 °C for 1 h (or over-night) in HDGN300 buffer (20 mM HEPES pH7,4, 300 mM KCl, 1 mM DTT, 10% glycerol, 1× complete, 0.05% NP40) to increase protein complexes solubilization. Isolation of insoluble (pellet) and soluble (supernatant) fractions was obtained via centrifugation (10,000× *g* for 10 min at 4 °C).

### 2.4. Immunoprecipitation (IP) of GFP-Tagged CYCLON

Magnetic nanobeads tagged with GFP (GFP-Trap^®^, Chromotek, Planegg, Germany) were prepared according to the manufacturer’s instructions. Soluble protein fraction was incubated with 30 µL of bead solution under rotation at 4 °C for 1 h. The flow-through fraction was kept for further analysis. The bound fraction was washed 3 times in HDGN300 before elution of bound proteins with 100 µL of Laemli buffer.

### 2.5. Immunoblotting

Proteins were loaded and separated on 4–12% Bis-Tris gel in MES buffer (Thermo Fisher) and transferred onto a PVDF membrane in Tris Glycine 10% ethanol for one hour at 100 V. Membranes were then treated with a blocking solution (PBS 0.1% Tween 20, 5% skimmed milk) before incubation with primary antibody (1/5000 α-GFP (Genentech, San Francisco, CA, USA), 1/1000 α-CYCLON (Bethyl laboratories, Montgomery, TX, USA) in PBS, 0.1% Tween 20, 5% skimmed milk), PBS 0.1% Tween washes and secondary antibody incubation (1/5000 anti-rabbit HRP (BioRad, Marne-La-Coquette, France). Revelation was performed through incubation in ECL SuperSignal West Pico Chemiluminescent Substrate (Thermo Fisher) and images captured using Vilber imaging system. The original blots are shown in the Appendix A.

### 2.6. Silver Staining

Protein samples were loaded and separated on a 4–12% NuPAGE™ Bis-Tris gel in MES buffer (Thermo Fisher Scientific). Silver staining of gel was performed according to the protocol for the SilverQuest silver staining kit provided by the manufacturer (Thermo Fisher Scientific).

### 2.7. Mass Spectrometry Analysis

Three independent replicates of CYCLON and control-IP fractions were stacked on the top of a 4–12% NuPAGE gel and stained with Coomassie blue R-250 before in-gel digestion using modified sequencing grade trypsin (Promega, Charbonnières-les-Bains, France) as previously described [16]. Resulting peptides were analyzed using online nanoliquid chromatography coupled to tandem MS (UltiMate 3000 and Q Exactive Plus, Thermo Fisher Scientific). Peptides were sampled on a 300 μm × 5 mm PepMap C18 precolumn and separated on a 75 μm × 250 mm C18 column (ReproSil-Pur 120 C18-AQ 1.9 μm) using a 240-min gradient. MS and MS/MS data were acquired using Xcalibur (Thermo Fisher Scientific). Peptides and proteins were identified using Mascot (version 2.6) through concomitant searches against Uniprot (Homo sapiens taxonomy, August 2018 version), classical contaminants database (homemade) and the corresponding reversed databases. Proline software [17] was used to filter the results: conservation of rank 1 peptides, peptide length ≥7 amino acids, false discovery rate (FDR) of peptide-spectrum match identification <1% as calculated on scores by employing the reverse database strategy, minimum peptide-spectrum match score of 25, and minimum of 1 specific peptide per identified protein group. Proline was then used to perform a compilation, grouping and MS1-based label free quantification of the protein groups from the different samples. Statistical analysis was performed using ProStaR [18]. Proteins identified in the reverse and contaminant databases, additional keratins and proteins quantified in less than 3 replicates of one condition were discarded. After log2 transformation, abundance values were normalized using the variance stabilizing normalization method before missing value imputation (slsa method for partially observed values in the condition and DetQuantile for data totally missing in one condition). Statistical testing was conducted using limma. Differentially expressed proteins were sorted out using a log2 (fold change) cut-off of 1.5 and a *p*-value ≤ 0.05, leading to an FDR < 5% according to the Pounds estimator.

### 2.8. Bioinformatics Analysis

The protein-protein interaction network was analyzed using Cytoscape 3.8 software [19]. Gene ontology analyses were performed using Funrich 3.1 [20] and DAVID resources. TCGA and GTEx data were mined using the GEPIA interface [21].

### 2.9. Immunofluorescence and Confocal Microscopy

1 × 10^6^ B593 GFP-CYCLON cells resuspended in PBS were plated on 12 mm glass coverslips (inside 24-well plates) for 30 min, fixed for 5 min in 4% PFA (Electron Microscopy Sciences), permeabilized in 0.2% Triton X-100 (Sigma) in PBS for 10 min and blocked in 3% bovine serum albumin (BSA, Sigma) in PBS (PBS 3% BSA) for 10 min. Cells were incubated with anti-NPM1 antibody (1:50, Santa Cruz, CA, USA) diluted in PBS 3% BSA for 1 h and washed three times in PBS. Goat anti-mouse secondary antibody coupled to AlexaFluor 546 (1:2000, Invitrogen) in PBS 3% BSA was added for 1 h and washed three times in PBS. Coverslips were mounted in DAPI-containing Mowiol (1:1000, EMD Milipore) solution overnight. Microscopy was performed on a confocal microscope (LSM 710, Zeiss Marly-le-Roi, France) equipped with a 63X/1.4 oil objective using the following settings: laser emission/detection (DAPI 405/405–491 nm; GFP 488/494–550 nm; RFP 561/566–684 nm), pinhole 0.7 µM, pixel dwell 3.14 µsec, speed 7, average 4, zoom 2.7×. Images were constructed using ZEN Blue software (Zeiss).

### 2.10. Patient Samples

This study was conducted in accordance with the Declaration of Helsinki and after obtaining the patients’ informed consent for participation in biological studies. A total of 97 DLBCL patients treated in the Grenoble Alpes University Hospital and Daniel Hollard Institute from January 2011 to December 2018 were included. The patient cohort was established based on the following inclusion criteria: de novo DLBCL tumors (*n* = 93), including EBV+ (*n* = 1) and HGBL (high-grade B-cell lymphoma, *n* = 3), but excluding all patients with previous history of lymphoma. Primary cutaneous DLBCL (leg type), primary DLBCL of the central nervous system and primary mediastinal large B-cell lymphoma were also excluded. IHC analysis of CYCLIN D1 and SOX11 were performed for CD5-positive DLBCL to exclude blastoid mantle cell lymphoma. We chose to include only patients treated with R-CHOP or R-CHOP-like in the first line (at least 3 cycles). Sufficient formalin-fixed paraffin-embedded (FFPE) (AFA fixation was excluded to avoid IHC bias) had to be available to build tissue microarray (TMA). Biological and clinical data were collected for each patient, including gender, age, IPI score, Ann Arbor stage, serum lactate dehydrogenase (LDH) level, therapeutic scheme as well as follow-up and survival data. DLBCLs were classified according to the last WHO classification into GC or non-GC subtype using the Hans algorithm. Additional details are given in Table 1.

### 2.11. Tissue Microarray (TMA) and Immunohistochemistry (IHC)

Tissue microarrays (TMAs) were built using four 0.6 mm cores (Alphametrix, Rödermark, Germany) from each FFPE DLBCL sample. Normal testis and appendix samples were also added respectively as internal positive and negative controls. TMA tissue spots were reviewed via morphological analysis (haematoxylin–eosin–saffron-stained section) to exclude those with an unrepresentative aspect (fibrotic or necrotic changes) and to ensure that every case showed morphology compatible with a DLBCL and a minimum of 90% of tumor cells. Immunohistochemical (IHC) analyses were performed on 4-μm sections of TMA blocks. A fully automated BenchMark ULTRA procedure (Roche Diagnostics, Meylan, France) was used (baking, deparaffinization, antigen retrieval, IHC and counterstaining). After a 64 min incubation in Ventana Cell Conditioning 1 retrieval solution (Tris/Borate/EDTA pH8), NPM1 mouse monoclonal antibody (NeoBiotech, Nanterre, France, clone 7H10B9) was incubated for 60 min at room temperature, at a 1:400 dilution. A Ventana Ultraview Universal DAB Detection Kit was used followed by Ventana Hematoxylin staining as a nuclear counterstain for 16 min. Specificity of staining was further confirmed with a second NPM1 mouse monoclonal antibody (Santa Cruz, Dallas, TX, USA, clone sc-271737) for 26 patients using a similar procedure except that non-specific binding and endogenous peroxidase were blocked using respectively 2% NGS solution and Discovery Inhibitor reagent. This antibody was incubated for 48 min at 36 °C, at a 1:400 dilution.

IHC staining was blindly scored by two different pathologists (AB and HS), using a scale from 1–4+ according to the percentage of positive cells (1+ [1–25%], 2+ [26–50%], 3+ [51–75%] and 4+ [76–100%]) and a scale from 1+ to 3+ according to the staining intensity (1+ weak, 2+ medium, 3+ strong). Discordant cases were the subject of joint proofreading. The reliability of positive cell rates and staining intensity scores between TMA cores was assessed using intraclass correlation (ICC) coefficients. For nuclear staining, the ICC values were 0.74 (95% CI: 0.66–0.81) for the percentage of positive cells and 0.71 (95% CI: 0.63–0.79) for staining intensity. For cytoplasmic staining, the ICC values were 0.79 (95% CI: 0.73–0.85) for the percentage of positive cells and 0.78 (95% CI: 0.72–0.85) for staining intensity. Collectively, ICC data demonstrated inter-core consistency for NPM1 staining.

### 2.12. Fluorescence In Situ Hybridization (FISH)

FISH was performed using 4 µm FFPE TMA sections. *MYC*, *BCL2* and *BCL6* rearrangements were evaluated using respectively a ZytoLight ^®^ SPEC MYC Dual Color Break Apart Probe, ZytoLight ^®^ SPEC BCL2 Dual Color Break Apart Probe and ZytoLight ^®^ SPEC BCL6 Dual Color Break Apart Probe (ZytoVision GmbH, Bremerhaven, Germany). A Vysis IGH/MYC/CEP 8 Tri-Color Dual Fusion Probe Kit (Abbott, Rungis, France) was used to determine if *IGH* was the partner gene of *MYC* rearrangement. At least 100 nuclei were analyzed for each sample. Nuclei were counterstained with DAPI/Vectashield^®^ (Vector Laboratories, Burlingame, CA, USA) and were analyzed with a Leica CytoVision GSL10 FISH fluorescence capture system^®^ (Leica, Wetzlar, Germany) using a 63× oil immersion objective. Signals were enumerated with the CytoVision imaging system^®^ (Leica). A cut-off value of 10% of the cells showing rearrangement was applied to consider a specimen positive.

### 2.13. Targeted NGS Sequencing

DNA was extracted from 3–4 FFPE core biopsies per case using a QIAsymphony DSP DNA Mini Kit (Qiagen, Les Ulis, France). DNA quality was assessed using TapeStation (Agilent, Les Ulis, France) and quantified using Qubit (Thermo Fisher Scientific). Targeted sequencing was performed using a capture approach covering 51 genes that present mutations in DLBCL, follicular lymphoma and chronic lymphocytic leukemia (*ARID1A, ATM, B2M, BCL2, BCL6, B-RAF, BORCS8-MEF2B, CARD11, CCND3, CD58, CD79A, CD79B, CDKN2A, CDKN2B, CDKN2B-AS1, CIITA, CREBBP, CXCR4, EP300, EZH2, FBXW7, FOXO1, GNA1, ID3, IRF4, ITPKB, KMT2A, KMT2D, KRAS, LOC100131635, MAL, MEF2B, MFHAS1, MYC, MYD88, NFKBIE, NOTCH1, NOTCH2, NRAS, PAX5, PIM1, PLCG2, PRDM1, REL, SOCS1, STAT6, TCF3, TNFAIP3,TNFRSF14, TP53* and *XPO1)*. This panel is based on the 34-genes panel developed for DLBCL by the French Lymphoma Study Association [22] that was extended to include additional genes of interest for diagnosis, prognosis and theranostic requirements in routine diagnostic workups of mature lymphoid neoplasms [23], as well as genes of interest for research purposes. Libraries were prepared using standard procedures using 50 ng of DNA and a KAPA HyperPlus kit (Roche). Libraries were purified, quantified and barcoded before being submitted to Illumina sequencing using a NextSeq 500/550 Mid Output Kit V2 (300 cycles) on a Nextseq 550 Illumina sequencer. Somatic mutation calling was performed using a custom sequence alignment, variant caller and annotation pipeline, as previously described [24]. Somatic mutations that affected protein coding regions (non-synonymous) and at exon splice-site junctions were retained for analysis. *KMT2D* analysis was not performed because of high background noise at this gene locus in sequencing data.

### 2.14. Statistics

Specific overall survival (SOS) and progression-free survival (PFS) were assessed using Kaplan–Meier analyses and Cox regression. The starting point for time-to-event analysis was the date of diagnosis. SOS was defined as the time to disease-related death and PFS as the time to disease progression with censoring at date of last contact. Patients without any event at the time of the last follow-up were censored. Median follow-up time was estimated using the reverse Kaplan–Meier method. The proportional-hazard assumption of the Cox models was checked based on Schoenfeld residuals after Cox model fitting. The fit of the Cox models was assessed using Harrell’s concordance index, which is defined as the probability that predictions and outcomes are concordant. Internal validation of the stability of the Cox models was assessed via bootstrap resampling. The prediction accuracy of multivariate models was evaluated using time-dependent receiver operating characteristic (ROC) curves and corresponding areas under time-dependent ROC curves (AUCs). Competing risk regression models were based on the Fine–Gray method [25]. All statistical analyses were performed using Stata 16.1 (Stata Corporation, College Station, TX, USA) or JPM 10 (SAS, Grégy-sur-Yerres, France).

## 3. Results

### 3.1. CYCLON Network Is Enriched in Nucleolar Proteins, Notably NPM1

To gain further insights into CYCLON oncogenic functions, we genetically engineered a DLBCL cell line (B593) to ectopically express endogenous levels of GFP-tagged CYCLON (Appendix A). Extraction of CYCLON was optimized via nucleic acid solubilization with MNase using chromatin digestion in mono- and di-nucleosomes as a read-out for reaction efficiency (Appendix A). GFP-CYCLON was then efficiently immunoprecipitated with GFP-TRAP nanobodies (Appendix A). Immunoprecipitated fractions were resolved using silver-stained SDS-PAGE (Appendix A) and subsequently analyzed using mass spectrometry-based label-free quantitative proteomics to identify CYCLON-associated proteins.

Mass spectrometry analyses revealed 32 proteins that specifically and reproducibly interact with CYCLON in this setting (Figure 1A, Appendix A). They include mainly nucleolar RNA-binding and ribosomal proteins, many of them being involved in gene expression regulation through RNA processing (Figure 1B). Several of these proteins are related to cell cycle progression and cell proliferation, involving at least partly MYC signaling (Figure 1A) and including centrosomal factors (Figure 1B), which is consistent with our previous results [10].

NPM1, DHX9 and HNRNPU lie at the crossroads of the different cellular functions represented in the CYCLON network. Of note, all these factors have been previously described as overexpressed in various solid or hematological cancers. NPM1 represents one of the most relevant candidates to be evaluated as a prognostic biomarker since it was shown to connect nucleolar function, MYC signaling [26] and cell division. Consistently with interatomic data, confocal microscopy in B593 GFP-CYCLON cells allowed nuclear co-localization between CYCLON and NPM1, mostly within, but also outside the nucleolus, to be confirmed (Figure 1C). The functional relationship between CYCLON and NPM1 is further supported by TCGA (The Cancer Genome Atlas) data that revealed than the two proteins are both overexpressed (Figure 1D) and showed a significant degree of co-expression in DLBCL (Figure 1E).

Oncogenic fusion proteins and gene mutations involving *NPM1* have been described respectively in anaplastic large-cell lymphoma (ALCL) [27] and acute myeloid leukemia (AML) [28,29]. However, *NPM1* mutations have not yet been identified in DLBCL.

### 3.2. NPM1 and CYCLON Co-Expression Is Predictive of R-CHOP Response in DLBCL Patients

To evaluate the clinical implication of NPM1 in DLBCL, NPM1 immunohistochemistry (IHC) was carried out on a representative cohort of 97 R-CHOP-treated DLBCL patients in which CYCLON expression was previously investigated [11] using tissue microarray (TMA). Clinical and pathological characteristics of this cohort are described in Table 1. In 23/97 (24%) cases results were negative for NPM1 staining (Appendix A, Table 1) and 3 cases were uninterpretable.

We first examined the co-expression of CYCLON and NPM1 in this series of clinical samples. We observed no strict dependency between CYCLON and NPM1 protein expression (Fisher exact test: *p* = 0.35), which in some cases could even be mutually exclusive (Figure 2A). However, 67 cases (71.2% of interpretable cases) presented a co-expression of CYCLON and NPM1. We then examined whether these double positive cases had a prognostic value in DLBCL.

Kaplan Meier survival analyses revealed that CYCLON/NPM1 double expressors DLBCL were significantly associated with an inferior outcome in terms of progression-free survival (PFS, Figure 2B, *p* = 0.021) compared to DLBCL expressing only CYCLON. The same tendency was observed for specific overall survival (SOS) without reaching statistical significance (Appendix A, *p* = 0.065).

We have recently shown on the same cohort that CYCLON displayed three distinct IHC staining patterns in DLBCL, namely diffuse pan-nuclear (pan), strictly nucleolar (nuc) or excluded from nucleolus (extra-nucleolar, ext), the latter being related to a far inferior prognosis. NPM1 positive cases did not correlate with any specific CYCLON staining pattern (Figure 2C; χ^2^ test: *p* = 0.15). Of note, we did not observe any NPM1-negative case presenting a CYCLON nucleolar pattern. Kaplan–Meier survival analyses allowed a group of 13 patients presenting both extra-nucleolar CYCLON [CYCLON ^(ext+)^] and positive NPM1 [NPM1^(+)^] staining associated with a high risk of presenting refractory disease or relapse to be further segregated (Figure 2D, *p* < 0.0001). This difference was also significant in term of SOS (Appendix A, *p* = 0.0005) and was also further supported by survival Cox model regression analyses that confirmed that CYCLON ^(ext+)^ represents a significant prognostic factor in terms of PFS and SOS only when co-expressed with NPM1 (Figure 2E). This first piece of information shows that co-expression of CYCLON and NPM1 is not neutral for prognosis compared to CYCLON expression alone and confirms the relevance of studying the co-expression of these two factors in DLBCL.

### 3.3. NPM1 Subcellular Localization Is Associated with Prognosis in DLBCL

A refined examination of NPM1 IHC positive cases showed three distinct IHC patterns (Figure 3A, Table 1): typical exclusive nuclear (47/97), exclusive cytosolic (10/97) or pan-cellular (both nuclear and cytosolic, 14/97). The reliability of these alternative NPM1 subcellular localizations was confirmed with a second monoclonal antibody directed against NPM1 (Appendix A). We then evaluated whether these different subgroups might impact DLBCL prognosis. Kaplan–Meier survival performed on all IHC subgroups revealed that an overall log-rank test was highly significant for both PFS (*p* = 0.0017, Figure 3B left) and SOS (*p* = 0.0081, Figure 3B right).

Strikingly, cases with a NPM1 pan-cellular pattern exhibited adverse outcomes (PFS: 24.8%, 95%-CI 4.8–52.6; SOS: 19%, 95%-CI 12.8–53.2) compared to patients showing negative (PFS: 84.3%, 95%-CI 58.9–94.7; SOS: 84.4% 95%-CI 59.1–94.7), exclusively nuclear (PFS: 49.9%, 95%-CI 24.3–71.1; SOS: 70.3%, 95%-CI 54.2–81.6%) or exclusively cytosolic staining (PFS: 46.7%, 95%-CI 15–73.7; SOS: 58.3%, 95%-CI 23–82.1). Consistently, a log rank test for linear trend across staining subtypes ordered as negative, exclusively nuclear, exclusively cytosolic and pan-cellular was also highly significant for PFS and SOS (*p* = 0.0002 and *p* = 0.002, respectively). Of note, these distinct NPM1 subcellular localizations seemed to be independent of the cell-of-origin classification (GC vs. non-GC, Appendix A; χ^2^ test: *p* = 0.54) and of any known genetic defects that could be evaluated here: recurrently mutated genes identified through a 51-gene targeted NGS lymphopanel (Appendix A) or *MYC* rearrangements (Appendix A). Similarly, no significant association was observed regarding mutational load (Appendix A), number of cytogenetic defects (Appendix A), or concomitant *MYC* and *BCL2*/*BCL6* rearrangements defining the HGBL subtype. The sample size prevented us from looking for any association of NPM1 staining pattern with newly described DLBLC subtypes [7,8].

The poor-prognosis pan-cellular NPM1 category (NPM1^(pan+)^: 14 cases) significantly differs from other patterns (nuclear, cytoplasmic, negative grouped within the category NPM1^(pan−)^: 80 cases) in Kaplan–Meier survival analyses (Appendix A) for both PFS (*p* = 0.001, left) and SOS (*p* = 0.007, right). Taken together, these data clearly demonstrate that NPM1 pan-cellular subcellular localization represents by itself a potent predictive factor for high-risk R/R DLBCL patients.

### 3.4. Multivariate Analysis of CYCLON, NPM1 and R-IPI on DLBCL Prognosis

We next investigated whether CYCLON and NPM1 could act as applicable prognostic biomarkers by adding to the R-IPI scoring. First, no relationship was found between CYCLON extra-nucleolar ^(+/−)^ and pan-cellular NPM1 ^(+/−)^ as shown using a standard χ^2^ test (*p* = 0.62). Next, multivariate survival Cox models for PFS and SOS were built with CYCLON extra-nucleolar cases ^(+/−)^, NPM1 pan-cellular cases ^(+/−)^ and R-IPI subdividing the patients into low-risk (R-IPI ^(low)^ = IPI score 0–2: 49 cases) and high-risk categories (R-IPI^(high)^ = IPI score 3–5: 48 cases). As shown in Table 2, CYCLON extra-nucleolar ^(+)^, NPM1 pan-cellular ^(+)^ and R-IPI ^(high)^ categories were statistically significant and independent prognostic factors with a strong negative impact on PFS and SOS.

The stability of the models was investigated using bootstrap resampling, providing 95% confidence intervals for hazard ratios that never include the value 1 for CYCLON, NPM1 and R-IPI variables in both PFS and SOS models. Finally, the Cox proportional hazard assumption was checked using the Schoenfeld residual test, and the discrimination ability of both models was satisfactory when assessed using Harrell’s C statistic with a concordance index of 0.74 for the PFS model and 0.76 for the SOS model. The prediction performance of the multivariate models did not depend markedly on follow-up time for PFS and SOS as assessed via ROC curves at 20, 40, 60, 80 and 90 months with an AUC in the range of 0.782–0.824 for PFS and 0.799–0.802 for SOS (Appendix A), indicating an end point-independent robust performance of PFS and SOS model predictions. This formally demonstrates that these atypical CYCLON and NPM1 localizations can, independently of R-IPI, predict relapse/refractory disease in DLBCL patients.

### 3.5. Multivariate Competing Risk Models Analyses Show That CYCLON, NPM1 and R-IPI Are Prognostic Markers of Refractory Disease-Related Death in DLBCL

Lastly, we examined whether extra-nucleolar CYCLON and pan-cellular NPM1 staining adjusted with the R-IPI score could specifically predict either refractory- disease or relapse-related death. To this end, competing risk (CR) regression models [25] were fitted, considering that refractory- and relapse-related death are competing events after R-CHOP treatment. CR analysis (Table 3) showed that extra-nucleolar CYCLON and pan-cellular NPM1 staining were strongly associated with an increased risk of refractory-related death (sHR = 4.04, *p* = 0.003; sHR = 4.64, *p* = 0.002, respectively).

In contrast, none of these staining patterns were retained as significant markers predicting relapse-related death (sHR = 1.08, *p* = 0.92; sHR = 1.63, *p* = 0.58, respectively). CR models also indicated that R-IPI was barely significant as a prognostic factor for refractory- or relapse-related death (sHR = 2.8, *p* = 0.043; sHR = 10.03, *p* = 0.043, respectively). Internal validation of the refractory-related death CR models assessed using a bootstrap 95% CI confirmed that all 3 sHRs for CYCLON, NPM1 and R-IPI factors were significantly different from 1. Extra-nucleolar CYCLON, pan-cellular NPM1 and high R-IPI score therefore represent independent prognostic factors that can specifically predict refractory-related death in R-CHOP-treated DLBCL.

## 4. Discussion

CYCLON was originally identified through a proteomic screen for “off-context” gene expression in a DBLCL cell line [10]. We described that this protein, mainly expressed in testis, was overexpressed in several B-cell lymphoma subtypes and associated with poor prognosis in R-CHOP-treated DLBCL [10,11]. Knock-down experiments showed that CYCLON controls a germline gene expression signature enriched in testis-specific factors and related to aggressive cancers [10]. CYCLON has been previously involved in models of B and T cells activation [12,13]. Of note, Dominguez-Sola et al. reported CYCLON induction upon MYC signaling activation during germinal-center B-cell selection [14]. Consistently, we also reported CYCLON as being, at least partly, a MYC downstream target in lymphoma cell lines [10]. Analysis of the CYCLON amino-acid sequence does not allow its potential molecular and biological functions to be accurately predicted. However, CYCLON is described as a highly phosphorylated protein with a C-terminal coil-coiled Cgr1-like domain, which has been involved in yeast nucleolar compartmentalization and ribosome biogenesis [30]. It also presents the interesting feature of being a fully intrinsically disordered protein. Altogether, this suggests that this protein might be involved in phase-separated nucleolar sub-structures.

The CYCLON interaction network derived from a B-cell line lymphoma cell line supports this hypothesis since it comprises mainly ribosomal and nucleolar proteins involved in RNA processing (Figure 1A,B). Interestingly, we also identified several proteins that have been related to MYC signaling and cell division, as well as centrosomal factors. These data are consistent with CYCLON physiological and pathological expressions in highly proliferating cells and suggest a role for this protein in cell cycle progression. Several proteins identified among the CYCLON interactome could be of interest as additional oncogenic factors contributing to its adverse effects in lymphoma cells.

We chose to further investigate NPM1, which is a ubiquitous, oligomeric and multi-domain protein that acts as a nucleolar organizer but has also been involved in multiple additional activities related to cell growth, homeostasis and stress response. NPM1 has a chaperone activity for ribosomal proteins and histones [31], as well as for transiently accumulated misfolded proteins in the nucleolus [32]. It has also been shown to regulate stability and localization of several tumor suppressor proteins including TP53 [33], thereby contributing to modulate apoptosis/cell proliferation balance. NPM1 has been further connected with the regulation of ribosome biogenesis and export, mitosis, replication, genome stability and transcription, which is consistent with the fact that NPM1 levels directly correlate with the protein synthesis rate and proliferation status of the cell [31]. Accordingly, NPM1 is overexpressed in highly proliferative cells and has been proven to be associated with several types of cancer. Oncogenic fusion proteins involving *NPM1* (*NPM1–ALK, NPM–RARα* or *NPM1–MLF1*) have also been described in ALCL [27] and in rare variants of AML [34]. Moreover, exon 12 *NPM1* gene mutations leading to NPM1 protein relocation in the cytoplasm (NPM1c) are found in 40% of AML [28,29]. NPM1-mutated AML have been recognized as a distinct entity in the last WHO myeloid neoplasm classification [34] related to favorable outcomes [26,29]. Interestingly, it has been shown that NPM1c leukemia express a distinctive stem cell-like gene expression pattern [35]. Moreover, *NPM1* mutations were found in committed progenitors and differentiated myeloid cells in AML but were absent from the stem cell and lymphoid compartments [36]. This suggests that NPM1c induce aberrant progenitor self-renewal in myeloid progenitors, which represents a critical step in the development of AML. Overall, the NPM1 effect in tumorigenesis appears to be complex and probably context-specific, varying amongst different tumor types. Indeed, NPM1 is thought to exert both oncogenic and tumor suppressor roles, since it has been described to be overexpressed, mutated, aberrantly located or even deleted in tumors [37]. Taken together, these observations prompted us to further investigate this protein in relation to CYCLON in DLBCL clinical samples.

IHC screening revealed that a combined expression of CYCLON and NPM1 in DLBCL was associated with an inferior prognosis in DLBCL (Figure 2B–D). However, the most intriguing observation was the distinct NPM1 subcellular localization patterns which had never been reported in this setting to our knowledge. Here, while 47 of 71 NPM1 positive cases presented the typical nuclear pattern (~65%), 10/71 (~15%) presented a cytosolic NPM1 localization and 14/71 (~20%) a pan-cellular NPM1 localization (Figure 3A). Although normally enriched in nucleoli, NPM1 is a “shuttling” protein, continuously travelling between the nucleolus, nucleoplasm and cytoplasm as required to fulfil its multiple functional roles. With the NPM1 wild-type form, this equilibrium favors the nuclear localization at steady state, which is observed in most normal and tumor tissues. As mentioned above, the NPM1 cytosolic form has exclusively been described in AML and in ALCL in relation with heterozygous mutations modifying NPM1 nuclear localization/export signals or specific fusion transcripts. However, no NPM1 mutation (deletion, insertion, substitution or CNV) has been reported to date in the literature or in the COSMIC database for the >1000 DLBCL cases for which NPM1 sequencing has been performed. Moreover, these mutations lead to a 100% cytosolic NPM1 relocation (which was also observed here in some DLBCL cases, whereas no such mutations have ever been described in B-NHL) but do not result in the specific mixed pan-cellular NPM1 staining pattern associated with poor prognosis, which has never been reported to date to our knowledge. This strongly suggests that the alternate NPM1 subcellular localization is not driven through any genetic defect but rather through alterations in protein or nucleic acid interactions (through cancer-related mislocalization of interacting partners for instance), protein-targeting signals, transport machinery or post-translational modifications [38].

As we initially identified NPM1 as a CYCLON interacting partner, we postulated that NPM1 could be involved in regulating CYCLON alternate sub-nuclear localization, for example, cytosolic NPM1 could be related to CYCLON exclusion from the nucleolus. As a matter of fact, it has been recently described that NPM1 is involved in nucleolar structuration through multiple proteins via R-tracts binding and rRNA interaction within liquid phase separation compartments [39]. CYCLON would represent a good candidate to participate in such functions as a disordered protein containing arginine-rich linear motifs. However, we were not able to show such an association by performing CYCLON and NPM1 IHC evaluation in DLBCL tissue sections, where staining intensity, number of positive cells and subcellular localization were strictly independent between both proteins.

The molecular network driving CYCLON exclusion from the nucleolus is most likely complex and regulated by post-translational modifications and multiple interactions involving both protein and nucleic acid interactions within phase-separated compartments, as previously reported for NPM1. This suggests that despite the fact that CYCLON and NPM1 can physically interact and show some degree of correlation in terms of gene expression in DLBCL cells, it seems that there is no strict co-regulation or functional association of the two proteins in DLBCL. However, we report here that concomitant expression of particular CYCLON and NPM1 IHC staining patterns can represent very potent predictors of refractory DLBCL (Figure 2B–D, Table 2).

We chose to favor interpretable, well-conducted internal bootstrap validation for estimating the generalizability of the models over poor, misconducted external validation based on small samples and/or unformal comparison between original and validation cohorts. This evaluation of model performance via bootstrapping fully supports the conclusion that CYCLON associated with the NPM1 IHC phenotype identifies patients with an inferior outcome. This, in our view, mitigates the necessity to validate in a larger cohort at this stage.

Several predictive biomarkers have been associated with DLBCL prognosis, before and after introduction of an immuno-chemotherapy regimen, and have been recently reviewed [40]. Their large number is consistent with the clinical and biological heterogeneity of DLBCL. However, besides R-IPI scoring, most genetic or molecular classifiers compatible with routine evaluation identified to date fail to accurately stratify all high-risk DLBCL patients and few of them are actually used in clinical practice. Here, we uncovered novel markers of R-CHOP-treated refractory DLBCL that rely on IHC staining of fixed tumor tissues, which is a standard automated procedure available in most centers involved in DLBCL management. Several protein markers are already routinely evaluated through IHC for DLBCL diagnosis (for instance CD10, BCL6 and MUM-1 used for the Hans algorithm). In this setting, CYCLON and NPM1 IHC staining interpretation would simply rely on the identification of their subcellular localization in DLBCL tumor cells, which can be easily distinguished by expert pathologists. A routine IHC detection of CYCLON and NPM1 could therefore be easily implemented for any suspected DLBCL and would help identification of high-risk patients.

Besides their prognosis impact, CYCLON and NPM1 could also have therapeutic implications to improve the clinical management of high-risk DLBCL. We have previously shown in DLBCL cell lines that CYCLON, as a downstream target of MYC, can be targeted by small molecule BET inhibitors, reducing CYCLON protein levels and mimicking the increased CD20 sensitivity associated with CYCLON knock-down [10]. CYCLON extra-nucleolar localization could be related to increased protein levels [11], suggesting that BET inhibition would be efficient to counteract the adverse effects related to this atypical subcellular localization. Of note, NPM1 has also been involved in the regulation of MYC signaling through binding both the MYC gene promoter and MYC protein. However, alternate NPM1 subcellular localizations have not been linked to increased gene expression/protein levels. Several therapeutic strategies have been explored to target aberrant cytosolic localization of mutant NPM1 in AML including inhibitors of nuclear export such as leptomycin B or selinexor [41,42], which has interestingly shown clinical activity in a phase I trial in refractory non-Hodgkin lymphoma [43]. Compounds targeting NPM1 oligomerization [44] or folding [45] have also been shown to induce apoptosis of AML cells expressing mutant NPM1. It is currently difficult to predict whether these strategies could be efficient in the context of the specific NPM1 pan-cellular localization associated with DLBCL poor prognosis.

In any case, rationalized therapeutic targeting of these unstructured and oligomeric proteins will require better understanding of the mechanisms driving their alternate subcellular localizations and identification among them of potential druggable pathways.

## 5. Conclusions

Overall, the present work represents the first report of the prognostic impact of NPM1 atypical localization in DLBCL and shows that CYCLON and NPM1 could cooperate in a clinical setting. The co-occurrence of extra-nucleolar CYCLON localization and a pan-cellular NPM1 pattern represents a strong predictor of poor prognosis in DLBCL, specifically associated with refractory disease. Their concomitant IHC evaluation using standard procedures would therefore represent an extremely potent way to identify high-risk primary refractory DBCL patients that could be readily implementable into clinical practice.

## Figures and Tables

**Figure 1 cancers-13-05900-f001:**
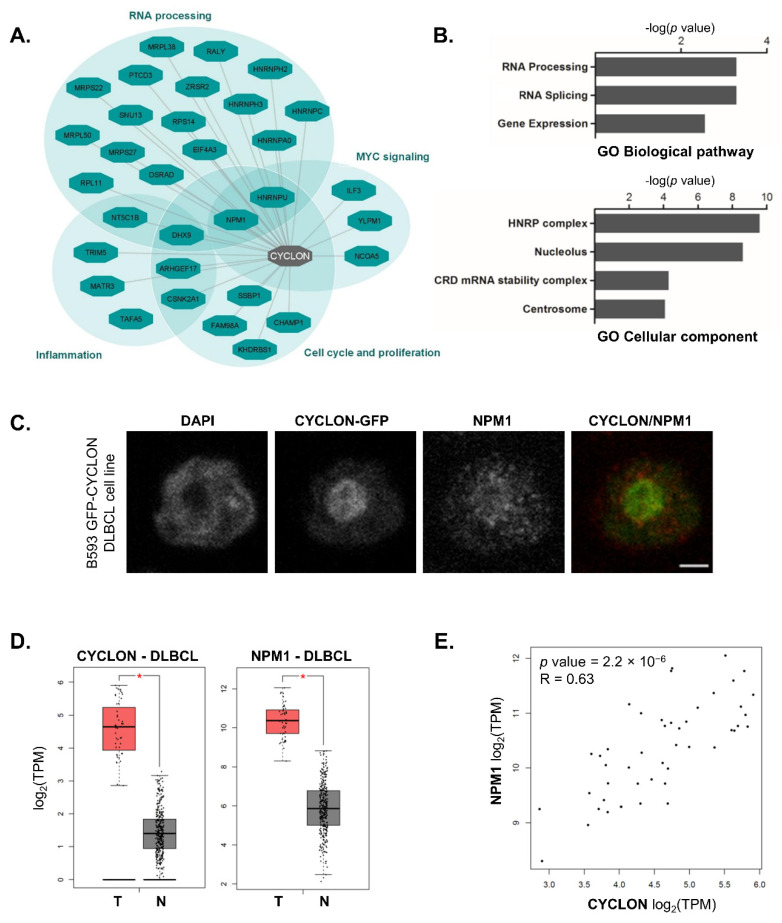
CYCLON protein network exerts oncogenic effects through nucleolar gene expression regulation and cell-cycle dependent functions. (**A**) CYCLON protein-protein interaction network and associated functions in B593 DLBCL cell line. (**B**) Gene ontology (GO) enrichment analysis of CYCLON interacting proteins for biological pathway (upper panel) and cellular component (lower panel) classifiers. (**C**) CYCLON and NPM1 sub-cellular interphase localization in B593 GFP-CYCLON cell line. Images were captured with a confocal microscope. Bar, 2 µm. (**D**) CYCLON and NPM1 gene expression levels in DLBCL TCGA samples (T) vs. normal B cells (N, from TCGA and GTEx databases), * *p* < 0.0001. (**E**) CYCLON and NPM1 gene expression correlation plot in DLBCL TCGA samples, R: Pearson coefficient. TCGA and GTEx data were retrieved through GEPIA interface.

**Figure 2 cancers-13-05900-f002:**
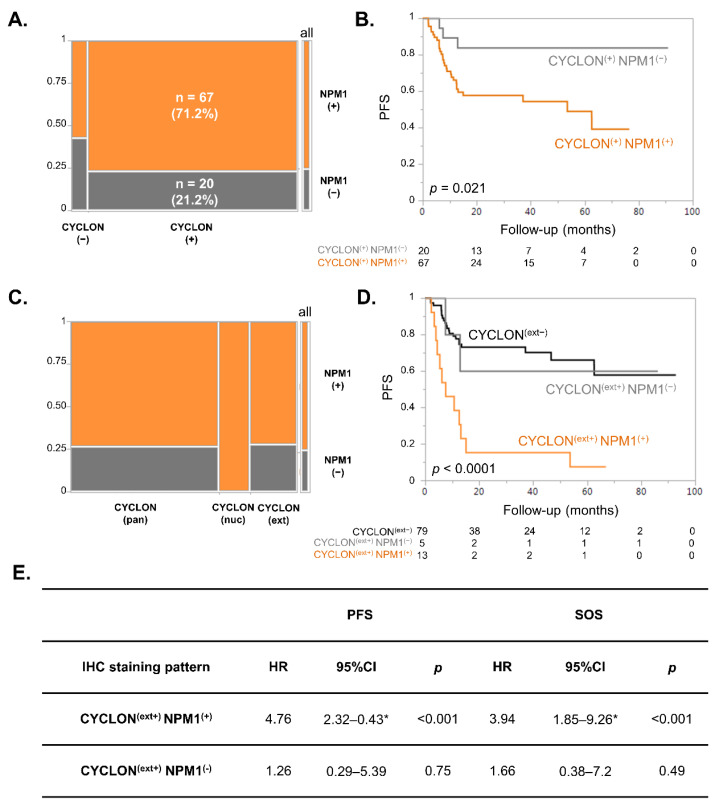
NPM1 and CYCLON co-expression is associated with progression-free survival in DLBCL patients. (**A**) Mosaic plot showing the distribution of NPM1^(−/+)^ cases according to CYCLON ^(−/+)^ status. (**B**) Kaplan–Meier analysis of progression-free survival (PFS) associated with CYCLON single expressors (CYCLON^(+)^ NPM1^(−)^) or CYCLON/NPM1 double expressors (CYCLON ^(+)^ NPM1^(+)^). (**C**) Mosaic plot showing the distribution of NPM1^(−/+)^ cases according to CYCLON IHC staining patterns (pan: pan-nuclear, nuc: nucleolar, ext: extra-nucleolar). (**D**) Kaplan–Meier analysis of PFS associated with CYCLON non-extra-nucleolar (CYCLON ^(ext−)^), CYCLON extra-nucleolar/NPM1 negative (CYCLON ^(ext+)^ NPM1^(−)^) or CYCLON extra-nucleolar/NPM1 double expressors (CYCLON ^(ext+)^ NPM1^(+)^). *p* values are derived from a log rank test. (**E**) Survival Cox model regression analyses of CYCLON ^(ext+)^ NPM1^(+)^ cases (n = 13) and CYCLON ^(ext+)^ NPM1^(−)^ cases (n = 5) for PFS and specific overall survival (SOS). CYCLON non-extra-nucleolar staining (CYCLON ^(ext−)^) (n = 79) is the reference category in both models. HR: hazard ratio, CI: confidence interval, * 95% CI based on bootstrap resampling (1000 replicates). Schoenfeld residual test: PFS model: global *p* = 0.91; SOS model: global *p* = 0.98.

**Figure 3 cancers-13-05900-f003:**
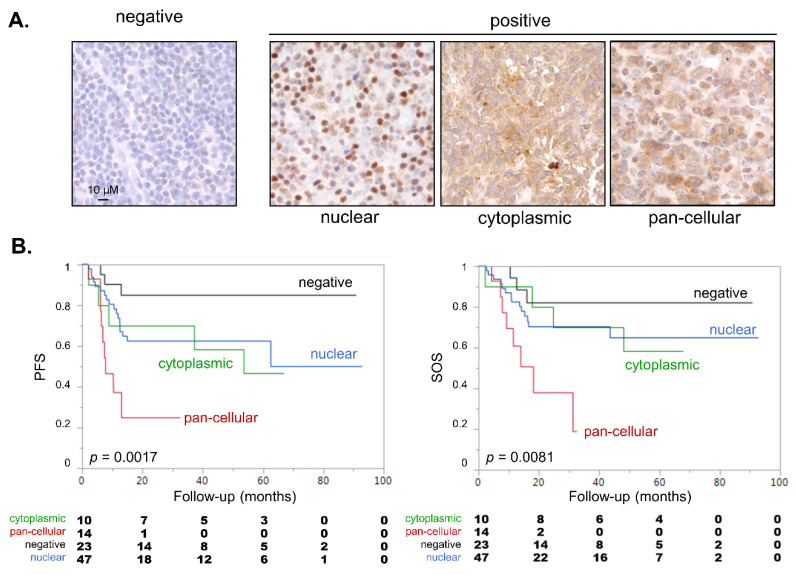
NPM1 subcellular localization is a predictive factor in DLBCL patients. (**A**) IHC analysis of NPM1 revealing distinct expression patterns in DLBCL patients as indicated. (**B**). Kaplan–Meier survival analysis of PFS (left) and SOS (right) associated with NPM1 IHC patterns. *p* values are derived from an overall log rank test.

**Table 1 cancers-13-05900-t001:** Cohort description.

Parameters	Value (Range)/% [Proportion]
Median age at diagnosis (years)	67 (29–94)
>60 years	69.1% [67/97]
Gender	
Male	55.7% [54/97]
Female	44.3% [43/97]
Median follow up time (months)	41.1 (2–93, 95% CI: 29.0–53.6)
First-line therapy	
R-CHOP	80.4% [78/97]
R-CHOP like	19.6% [19/97]
Mini R-CHOP	8.2% [8/97]
R-CVP	9.3% [9/97]
R-CHVP	1.0% [1/97]
R-COP	1.0% [1/97]
Complete response rate	78.3% [76/97]
Overall response rate	83.5% [81/97]
Primary refractory cases	20.6% [20/97]
Relapse cases	18.5% [18/97]
Specific overall survival (SOS)	62.0% (95% CI: 49.4–72.4)
Progression-free survival (PFS)	51.1% (95% CI: 36.2–64.1)
Ann Arbor staging classification (diagnosis)	
I (Single lymph node (LN) involved)	11.3% [11/97]
II (2 or more LN ipsilateral to the diaphragm)	20.6% [20/97]
III (LN on both sides of the diaphragm)	21.6% [21/97]
IV (extralymphatic organs or tissues involvment)	46.4% [45/97]
LDH > upper limit of normal	75.0% [73/97]
NPM1 (IHC)	
Negative	23.7% [23/97]
Cytoplasmic	10.3% [10/97]
Nuclear	48.4% [47/97]
Pan-cellular	14.4% [14/97]
Uninterpretable	3.2% [3/97]

**Table 2 cancers-13-05900-t002:** Multivariate bootstrap Cox regression analysis of the significance of independent prognostic variables NPM1, CYCLON and R-IPI. HR: hazard ratio; R-IPI: revised International Prognostic Index, 95% CI: confidence intervals based on bootstrap resampling (1000 replicates, SE: standard error on HR computed using the default method; 3 cases were uninterpretable for NPM1 staining. Schoenfeld residual test: PFS model: global *p* = 0.74; SOS model: global *p* = 0.64; Harrell’s C statistic: PFS model: C = 0.74; SOS model: C = 0.76. NPM1 pan-cellular ^(+)^: *n* = 14, pan-cellular^(−)^: *n* = 83, CYCLON extra-nucleolar^(+)^: *n* = 18, extra-nucleolar^(−)^: *n* = 79, R-IPI^(high)^: *n* = 48, R-IPI^(low)^: *n* = 49.

Variable	Category	HR	95% CI	*p*-Value	SE	HR	95% CI	*p*-Value	SE
NPM1	pan-cellular^(+)^ versus pan-cellular^(−)^	5.2	2.1–12.6	<0.001	2.24	5.8	2.5–13.3	<0.001	1.07
CYCLON	extra-nucleolar^(+)^ versus extra-nucleolar^(−)^	2.9	1.4–6.2	0.002	1.03	2.8	1.2–6.8	0.007	2.67
R-IPI	R-IPI^(high)^ versus R-IPI ^(low)^	3.7	1.5–8.9	0.001	1.46	5.2	2.0–13.6	<0.001	2.38

**Table 3 cancers-13-05900-t003:** Competing risk (CR) regression models identifying IHC staining patterns associated with refractory- or relapse-related death. ^1^ Event of interest: refractory-related death, competing event: relapse-related death, ^2^ event of interest: relapse-related death, competing event: refractory-related death, ^3^ subhazard ratio, ^4^ confidence interval based on robust variance estimates, ^5^ confidence interval based on bootstrap resampling (500 replicates), ^6^ CYCLON extra-nucleolar ^(+)^ versus CYCLON extra-nucleolar ^(−)^ staining, ^7^ NPM1 pan-cellular ^(+)^ versus NPM1 pan-cellular ^(−)^, ^8^ R-IPI high versus low score.

Competing Risk (CR)	Refractory ^1^				Relapse ^2^		
	sHR ^3^	95%CI ^4^	*p*	Bs-95%CI ^5^	sHR	95%CI ^4^	*p*
CYCLON ^6^	4.04	1.62–10.05	0.003	1.32–12.09	1.08	0.24–4.92	0.92
NPM1 ^7^	4.64	1.79–12.04	0.002	1.08–14.46	1.63	0.29–9.29	0.58
R-IPI ^8^	2.8	1.03–7.59	0.043	1.008–10.24	10.03	1.07–94.6	0.043

## Data Availability

The data presented in this study are available in Appendix A.

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
