# Peer review of "CYCLON and NPM1 Cooperate within an Oncogenic Network Predictive of R-CHOP Response in DLBCL"

_cancers, 2021, doi:10.3390/cancers13235900_

Round 1
Reviewer 1 Report
The authors have answered and processed the comments extensively and carefully and have thus significantly improved the manuscript. As indicated, a relatively small DLBCL cohort (n<100) is studied, which has also already been published with another biomarker. Unfortunately, the biggest concern therefore remains, namely the lack of a validation cohort. Despite extensive statistical analysis with bootstrap methodology, it is necessary to validate the results in a large independent cohort. Certainly because it is doable with (inter)national contacts to be able to include these validation results in the not too long term. After all, it only concerns performing IHC in association with clinical outcomes in a well-defined DLBCL cohort.
Author Response
First we would like to thank the reviewer for his positive comments on the revised version of our manuscript.
From a theoretical viewpoint, we agree that external validation in an independent cohort is an important aspect to assess the reliability of predictive modeling. However, in practice, this point is debatable, since external validation is not free of bias unless carefully performed. Various aspects may differ between original and validation samples, including clinical staging, biological parameters, age, mutational profiles, treatment protocols together with several other important variables in DLBCL. Validation samples should therefore have features plausibly related to the original samples, but at the same time, be divergent enough to ensure the generalizability of the model. Differences between original and validation cohorts for variables not included in the model, termed “missed predictors” (as examples: age, Ann-Arbor staging, MYC/TP53 gene alterations together with many other crucial parameters) can affect model coefficients (and therefore Hazard Ratios) and seriously impact on the validity of the external validation process of the CYCLON/NPM1 IHC staining-based model and related conclusions.
Most importantly, validation results based on small samples can be biased due to systematic differences, confounding and correlation effects, leading to unreliable comparison with the original cohort. This can be checked by C-statistics, calibration and more or less complicated methods specifically intended to time-to-event outcome. Therefore, only large cohort including high number of events can ensure reliable conclusions based on external validation. To have reasonable power for formal comparison of models between original and validation samples, expert in the field considers that at least 100 events (but preferably 250 events) are needed for external validation (1). Because NPM1 pan-cellular staining and CYCLON extra-nucleolar staining represent approximately 15-20% of cases (events), validation cohort should include at least 500 cases, and certainly more considering that an “event” for these IHC staining categories is “relapse” or “death”.
As mentioned previously, despite all our efforts, we were not able to have access to such a large cohort with proper clinical annotations within our collaborative network. Because formal external validation is clearly not feasible in a reasonable time, we focused on internal validation, which is a crucial step to evaluate model performance. Bootstrapping was performed at each step of this work, and in the revised manuscript we have now added bootstrap-based 95%CI of HR in Figure 2E, indicating the robustness of the conclusion that CYCLON(ext+) associated with NPM1(+) IHC phenotype identifies patients with inferior outcome.
Overall, we chose to favor interpretable, well-conducted internal bootstrap validation for estimating the generalizability of the models over poor, misconducted external validation based on small samples or/and unformal comparison between original and validation cohorts, as it appears in too many papers.
We hope that you will find this comment convincing.
- Steyerberg EW. Clinical prediction models. A practical approach to development, validation, and updating. Chapter 17. Validation of prediction models, pp 299-310. Chapter 19. Patterns of external validity, pp 333-358. Springer, 2009.
Reviewer 2 Report
The authors have addressed my concerns mainly in the Discussion.
Author Response
We would like to express our sincere thanks to the reviewer for his in-depth evaluation of our manuscript and his constructive comments and suggestions, which have been very helpful to improve and strengthen our report.
This manuscript is a resubmission of an earlier submission. The following is a list of the peer review reports and author responses from that submission.
Round 1
Reviewer 1 Report
Bouroumeau and colleagues evaluated the expression and subcellular localization of CYCLON and NPM1 in a cohort of DLBCL patients. The study is well designed and addresses the relevant topic of prognostication in DLBCL, a highly heterogeneous disease in need of more accurate prognostic systems. Unlike R-IPI, which is based on clinical variables, the authors identify biology-based predictors for R-CHOP response.
- Line 66. To my knowledge, the ABC, and not the GCB, subtype is associated with an inferior outcome.
- It would be valuable to expand the discussion by giving some clues about how CYCLON and NPM1 might be targeted therapeutically in DLBCL.
- Are CYCLON+NPM1+ double expressor cases at increased risk for CNS recurrence?
- Did the authors analyze CYCLON and NPM1 expression and subcellular localization in transformed DLBCL? Is CYCLON and NPM1 expression increased from low grade to high grade histology?
Author Response
Reply to Reviewer 1
Bouroumeau and colleagues evaluated the expression and subcellular localization of CYCLON and NPM1 in a cohort of DLBCL patients. The study is well designed and addresses the relevant topic of prognostication in DLBCL, a highly heterogeneous disease in need of more accurate prognostic systems. Unlike R-IPI, which is based on clinical variables, the authors identify biology-based predictors for R-CHOP response.
- Line 66. To my knowledge, the ABC, and not the GCB, subtype is associated with an inferior outcome.
We sincerely apologize for that mistake, which has been corrected in the manuscript.
- It would be valuable to expand the discussion by giving some clues about how CYCLON and NPM1 might be targeted therapeutically in DLBCL.
We have previously shown in DLBCL cell lines that CYCLON, as a downstream target of MYC, can be targeted by BET inhibitors, reducing CYCLON protein levels and mimicking the increased CD20 sensitivity associated with CYCLON knock-down [1]. CYCLON extra-nucleolar localization seems to be related to increased protein levels [2], suggesting that BET inhibition could be efficient to counteract the adverse effects associated with this atypical subcellular localization. Of note, NPM1 has also been involved in regulation of MYC signaling through binding both MYC gene promoter and MYC protein.
However, alternate NPM1 subcellular localizations have not been associated with increased gene expression / protein levels.
Several therapeutic strategies have been explored to target aberrant cytosolic localization of mutant NPM1 in AML including inhibitors of nuclear export such as Leptomycin B or Selinexor [3] [4], which has interestingly shown clinical activity in a phase I trial in refractory non-Hodgkin lymphoma [5]. Compounds targeting NPM1 oligomerization [6] or folding [7] have also been shown to induce apoptosis of AML cells expressing mutant NPM1. But, it is difficult to predict whether these strategies could be efficient in the context of the specific NPM1 pan-cellular localization associated with DLBCL poor prognosis.
In any case, rationalized therapeutic targeting of these unstructured and oligomeric proteins will require to better understand the mechanisms driving their alternate subcellular localizations and identify among them potential druggable pathways.
The above section has been integrated to the discussion part of the manuscript.
- Are CYCLON+NPM1+ double expressor cases at increased risk for CNS recurrence?
In our cohort, only 2 cases presented a CNS relapse, which is consistent with the reported incidence of such events (2-5% in DLBCL). Of note, both cases were actually double positive for CYCLON and NPM1, presenting respectively extra-nucleolar CYCLON/nuclear NPM1 and pan-nuclear CYCLON/pan-cellular NPM1. However, such small sample size does not allow to formally conclude on any association between NPM1/CYCLON positivity or specific staining and CNS recurrence.
- Did the authors analyze CYCLON and NPM1 expression and subcellular localization in transformed DLBCL? Is CYCLON and NPM1 expression increased from low grade to high grade histology?
We would like to thank the reviewer for this very interesting question. As indicated in the section detailing inclusion criteria, this study was designed to focus on de novo DLBCL, so we did not experimentally evaluate CYCLON and NPM1 localizations in transformed DLBCL. Public data mining performed across non-Hodgkin B-lymphoma (B-NHL) subtypes revealed an increased gene expression of CYCLON in aggressive B-NHL subtypes compared with indolent forms [1], suggesting an association with disease aggressiveness. However, no such profile was observed for NPM1 (please see attachment), whose expression does not vary across subtypes.
However, here again, it is important to keep in mind that the above data reflect only NPM1 gene expression and do not imply changes in NPM1 subcellular localization. Only a detailed exploration of CYCLON and NPM1 localization across low and high grade B-NHL will allow to definitely answer the question.
References :
- Emadali, A.; Rousseaux, S.; Bruder-Costa, J.; Rome, C.; Duley, S.; Hamaidia, S.; Betton, P.; Debernardi, A.; Leroux, D.; Bernay, B.; et al. Identification of a novel BET bromodomain inhibitor-sensitive, gene regulatory circuit that controls Rituximab response and tumour growth in aggressive lymphoid cancers. EMBO molecular medicine 2013, 5, 1180-1195, doi:10.1002/emmm.201202034.
- Bouroumeau, A.; Bussot, L.; Sartelet, H.; Fournier, C.; Betton-Fraisse, P.; Col, E.; David-Boudet, L.; McLeer, A.; Lefebvre, C.; Raskovalova, T.; et al. Extranucleolar CYCLON Staining Pattern Is Strongly Associated to Relapse/Refractory Disease in R-CHOP-treated DLBCL. HemaSphere 2021, 5, e598, doi:10.1097/HS9.0000000000000598.
- Etchin, J.; Sanda, T.; Mansour, M.R.; Kentsis, A.; Montero, J.; Le, B.T.; Christie, A.L.; McCauley, D.; Rodig, S.J.; Kauffman, M.; et al. KPT-330 inhibitor of CRM1 (XPO1)-mediated nuclear export has selective anti-leukaemic activity in preclinical models of T-cell acute lymphoblastic leukaemia and acute myeloid leukaemia. British journal of haematology 2013, 161, 117-127, doi:10.1111/bjh.12231.
- Garzon, R.; Savona, M.; Baz, R.; Andreeff, M.; Gabrail, N.; Gutierrez, M.; Savoie, L.; Mau-Sorensen, P.M.; Wagner-Johnston, N.; Yee, K.; et al. A phase 1 clinical trial of single-agent selinexor in acute myeloid leukemia. Blood 2017, 129, 3165-3174, doi:10.1182/blood-2016-11-750158.
- Kuruvilla, J.; Savona, M.; Baz, R.; Mau-Sorensen, P.M.; Gabrail, N.; Garzon, R.; Stone, R.; Wang, M.; Savoie, L.; Martin, P.; et al. Selective inhibition of nuclear export with selinexor in patients with non-Hodgkin lymphoma. Blood 2017, 129, 3175-3183, doi:10.1182/blood-2016-11-750174.
- Balusu, R.; Fiskus, W.; Rao, R.; Chong, D.G.; Nalluri, S.; Mudunuru, U.; Ma, H.; Chen, L.; Venkannagari, S.; Ha, K.; et al. Targeting levels or oligomerization of nucleophosmin 1 induces differentiation and loss of survival of human AML cells with mutant NPM1. Blood 2011, 118, 3096-3106, doi:10.1182/blood-2010-09-309674.
- Urbaneja, M.A.; Skjaerven, L.; Aubi, O.; Underhaug, J.; Lopez, D.J.; Arregi, I.; Alonso-Marino, M.; Cuevas, A.; Rodriguez, J.A.; Martinez, A.; et al. Conformational stabilization as a strategy to prevent nucleophosmin mislocalization in leukemia. Scientific reports 2017, 7, 13959, doi:10.1038/s41598-017-14497-4.
- Basso, K.; Margolin, A.A.; Stolovitzky, G.; Klein, U.; Dalla-Favera, R.; Califano, A. Reverse engineering of regulatory networks in human B cells. Nature genetics 2005, 37, 382-390, doi:10.1038/ng1532.

Reviewer 2 Report
The work described by Bouroumeau et al. regarding in the prognostic impact of NPM1 in RCHOP-treated DLBCL patients is interesting. To our knowledge, studying the role of NPM1 in DLBCL is a novelty. The method used is descriptive and not with any approach to causality. Although the laboratory results show a clear correlation between CYCLON and NPM1, the clinical data are not well substantiated. Therefore, we have major concerns publishing the manuscript in its current format.
Major comments:
- From a methodological point of view, an independent validation cohort, including whole slides NPM1 evaluation, is indispensable. Only a relatively small DLBCL cohort has been studied (n<100), which has already been published demonstrating the role of CYCLON.
- From a statistical point of view, using survival modelling, given the relatively low number of events (~40), it is not suitable to evaluate more than 4 variables, as currently CYCLON, NPM1, cellularity of expression, R-IPI etc. have been studied. Results would be more convincing in a cohort with twice the current numbers.
- There is no clear description of the group of DLBCL cases included and does not conform to the latest WHO classification. Specify diagnoses, like DLBCL-NOS, HGBCL’s, EBV positivity, etc. Are there differences in positivity in NPM1 or CYCLON among these sub-entities?
- The quality of the IHC analysis could be improved, since 1) there is a lot of background staining and 2) analysis has been performed on TMAs instead of whole slides. Besides, the tumor cell percentage, kappa values and ICC scoring are missing which are important in indicating the quality of the IHC analysis.
- The effect of different treatments or DLBCL subtypes have not been evaluated in a multivariate analysis, f.e. anthracycline based therapy (RCHOP) versus others, etc.
- The discussion section is mainly focused on CYCLON, which was recently published. Balance this section with more content regarding NPM1, the novelty of this study.
- The interpretation of the results is throughout the manuscript too firmly formulated, f.e. by stating that this IHC evaluation could readily be implemented into clinical practice and a critical evaluation based on other literature is missing in the discussion.
Minor comments:
- Although a strong gene expression correlation in the laboratory results, there is no association between IHC CYCLON and NPM1. Can the authors explain this?
- Besides this correlation, some cases are CYCLON negative, but positive for NPM1. Can the authors comment on this observation?
- There are several minor language issues, please check the manuscript carefully.
- Please avoid the frequent use of superlatives.
- The introduction stated that the GCB subtype corresponds with an inferior survival while ABC subtype corresponds with inferior survival.
- The original referral articles referencing COO-classification are lacking
- DHX9 and HNRNPU at page 13 (sentence 401) were not mentioned anywhere else in the article.
- For readability, all figures should be greatly improved in editing
- Please add the numbers at risk to the KM-curves
Author Response
Reply to Reviewer 2
The work described by Bouroumeau et al. regarding in the prognostic impact of NPM1 in RCHOP-treated DLBCL patients is interesting. To our knowledge, studying the role of NPM1 in DLBCL is a novelty. The method used is descriptive and not with any approach to causality. Although the laboratory results show a clear correlation between CYCLON and NPM1, the clinical data are not well substantiated. Therefore, we have major concerns publishing the manuscript in its current format.
Major comments:
- From a methodological point of view, an independent validation cohort, including whole slides NPM1 evaluation, is indispensable. Only a relatively small DLBCL cohort has been studied (n<100), which has already been published demonstrating the role of CYCLON.
Despite all our efforts, it was not possible to have access to more samples in reasonable time and cost to increase our cohort size or provide a validation set. However, internal validation by bootstrap analysis of Cox regression model performed for multivariate models is robust and confirmed the validity of the conclusions derived from this observational retrospective study. This, in our view, mitigates the necessity to validate in a larger cohort at this stage. We are really willing to have the opportunity to have access to larger DLBCL series to further confirm both CYCLON and NPM1 predictive accuracy. This analysis should be planned with the French Lymphoma Study Association (LYSA group).
Regarding IHC evaluation on whole slides, we performed several controls to make sure that TMA tissue sections were representative of whole tissue sections. First, for each antibody tested, IHC experimental set-ups were performed on DLBCL complete sections, as well as control tissues, and allowed to demonstrate that the staining was specific and homogeneous over the whole sections.
Then, TMAs were built to ensure an optimal representativeness of the initial samples: for each case, 4 core biopsies were collected in distinct areas presenting >90% of DLBCL tumor and avoiding necrosis or fibrosis areas. We believe that this strategy is valid to avoid potential technical or interpretation biases and guarantee an optimal reliability in the results.
- From a statistical point of view, using survival modelling, given the relatively low number of events (~40), it is not suitable to evaluate more than 4 variables, as currently CYCLON, NPM1, cellularity of expression, R-IPI etc. have been studied. Results would be more convincing in a cohort with twice the current numbers.
We agree that it is usually admitted that a minimum of 30 cases per variable is required to obtain valuable estimates and standard errors (SE) of model coefficients. However, this can be mitigated depending on the experimental design. With only few cases per variable, the theoretical risk with multivariate models is overfitting, that results in high standard errors of Hazard Ratios (HRs). This is not the case in the multivariate Cox model presented in Table 2: we have added to the table standard errors of HRs computed by the default method (default SE), which remain at reasonable values for all three variables (CYCLON, NPM1 and R-IPI), contributing to the high statistical significance of HRs (p=0.007). Moreover, 95% CI of HRs based on bootstrap resampling are not excessively large (extreme values: 1.2 to 13.6) and never include the value 1 as possible HR, supporting that the multivariate Cox model and related conclusions are acceptable despite the limited size and number of events of this cohort.
- There is no clear description of the group of DLBCL cases included and does not conform to the latest WHO classification. Specify diagnoses, like DLBCL-NOS, HGBCL’s, EBV positivity, etc. Are there differences in positivity in NPM1 or CYCLON among these sub-entities?
We thank the reviewer for pointing at this imprecision in our manuscript: we chose to include in our cohort de novo DLBCL NOS (n=93), as well as EBV+ (n=1) and HGBL (n=3). Other atypical DLBCL presentations including primary cutaneous DLBCL - leg type, primary DLBCL of the central nervous system and primary mediastinal large B-cell lymphoma were excluded. This information is now reported in the Material and Methods / Patient samples section.
The EBV+ case presented a pan-nuclear CYCLON staining and a cytosolic NPM1 staining. The 3 HGBL cases also presented a pan-nuclear CYCLON staining; two of them were negative for NPM1 whereas the third one presented a nuclear form of NPM1. However, the small sample size does not allow to formally conclude on any association between NPM1/CYCLON positivity or specific staining and these sub-entities.
- The quality of the IHC analysis could be improved, since 1) there is a lot of background staining and 2) analysis has been performed on TMAs instead of whole slides. Besides, the tumor cell percentage, kappa values and ICC scoring are missing which are important in indicating the quality of the IHC analysis.
IHC pictures presented in Figure 3A have been replaced by higher resolution images. We hope that they are now of better quality.
As developed above (point 1), we made sure to limit as much as possible any bias associated with TMAs. DLBCL are particularly adapted for this strategy because composed mostly of tumor cells. Then, each of the 4 replicate biopsies per tissue section/case included in the TMAs was collected from distinct tumor-rich areas (>90% tumor cells, excluding fibrotic/necrotic areas) to ensure representativeness between the initial tumor and the TMAs. Every IHC staining was double-blindly read by two different pathologists and the few discordant cases were reviewed jointly.
To further confirm the robustness of our approach, as suggested by the reviewer, reliability of positive cell rates and staining intensity scores between TMA cores was assessed using intra-class correlation (ICC) coefficients. For nuclear staining, the ICC values were 0.74 (95%CI: 0.66-0.81) for the percentage of positive cells and 0.71 (95%CI: 0.63-0.79) for staining intensity. For cytoplasmic staining, the ICC values were 0.79 (95%CI: 0.73-0.85) for the percentage of positive cells and 0.78 (95%CI: 0.72-0.85) for staining intensity. Collectively, the ICC data demonstrated inter-core consistency for NPM1 staining.
This paragraph has been added at the end of the Material and Methods / TMA-IHC section.
- The effect of different treatments or DLBCL subtypes have not been evaluated in a multivariate analysis, f.e. anthracycline based therapy (RCHOP) versus others, etc.
We would like to thank the reviewer for this interesting suggestion. A small number of patients (10/97, 10.3%) were treated by regimen not including anthracycline (R-CVP: 9 cases, R-COP: 1 case). Univariate survival cox models comparing anthracycline status of regimen showed that this category of patients had increased risk of relapse or death in comparison with patients treated by R-CHOP (PFS: HR=4.07, 95%CI: 1.77-9.36; p=0.001; SOS: HR=5.92, 95%CI: 2.49-14.03; p<0.001, please see attachment).
Standard chi-squared tests did not reveal any association between treatment with or without anthracycline and CYCLON extra-nucleolar (+/-) staining (p=0.90) and NPM1 pan-cellular (+/-) staining (p=0.52). Moreover, the presence/absence of anthracycline in the regimen did not affect the effect of CYCLON extra-nucleolar (+/-) or NPM1 pan-cellular (+/-) effects on PFS and SOS as demonstrated in Cox regression models where the included interaction terms between CYCLON and NPM1 with anthracycline status did not reach statistical significance (PFS: p= 0.18 for CYCLON*anthracycline interaction term; p=0.73 for NPM1*anthracycline interaction term; SOS: p= 0.15 for CYCLON*anthracycline interaction term; p=0.91 for NPM1*anthracycline interaction term).
These data demonstrated that CYCLON extra-nucleolar (+/-) staining, NPM1 pan-cellular (+/-) staining and anthracycline (+/-) status can be considered as statistically and functionally independent prognostic factors.
Regarding the low number of cases with no anthracycline in the regimen, bootstrap confidence interval of model coefficient for this variable cannot be reliably estimated in multivariate cox models, and this is the reason why the variable “anthracycline (+/-)” was not included in the final multivariate models including NPM1, CYCLON and R-IPI (Table 2).
- The discussion section is mainly focused on CYCLON, which was recently published. Balance this section with more content regarding NPM1, the novelty of this study.
We have extended the discussion by adding several sections that further describe NPM1 cellular function and comment its potential involvement in cancer as well as the mechanisms that could drive its alternate localization in DLBCL (see also reply to reviewer’s 3 comments).
- The interpretation of the results is throughout the manuscript too firmly formulated, f.e. by stating that this IHC evaluation could readily be implemented into clinical practice and a critical evaluation based on other literature is missing in the discussion.
This suggestion was taken into account, and we moderated our interpretations and conclusions throughout the manuscript.
We would like to acknowledge the suggestion to provide a critical evaluation of current prognosis biomarker in DLBCL. However, we really believe in the implementability of CYCLON and NPM1 screening to evaluate DLBCL prognosis in the routine clinical practice. We propose to extend the discussion with the following paragraph:
“Several predictive biomarkers have been associated with DLBCL prognosis, before and after introduction of immuno-chemotherapy regimen, and have been recently reviewed (Papageorgiou, Therapeutic advances in hematology, 2021). Their large number is consistent with the clinical and biological heterogeneity of DLBCL. However, besides R-IPI scoring, most genetic or molecular classifiers compatible with routine evaluation identified to date fail to accurately stratify all high-risk DLBCL patients and few of them are currently used in the clinical practice.
Here, we uncovered novel markers of R-CHOP-treated refractory DLBCL that rely on IHC staining of fixed tumor tissue, which is a standard automated procedure available in most centers involved in DLBCL management. Several protein markers are already routinely evaluated through IHC for DLBCL diagnosis (for instance CD10, BCL6 and MUM-1 used for the Hans algorithm). In this setting, CYCLON and NPM1 IHC staining interpretation would simply rely on the identification of their subcellular localization in DLBCL tumor cells, which can be easily distinguished by expert pathologists. A routine IHC detection of CYCLON and NPM1 could therefore be easily implemented for any suspected DLBCL and would help identification of high-risk patients”.
Minor comments:
- Although a strong gene expression correlation in the laboratory results, there is no association between IHC CYCLON and NPM1. Can the authors explain this?
- Besides this correlation, some cases are CYCLON negative, but positive for NPM1. Can the authors comment on this observation?
In B593 DLBCL cell line in which CYCLON/NPM1 interaction was identified, GFP-CYCLON and NPM1 partly co-localize within the nucleolus. As mentioned by the reviewer, TCGA data mining reveals both overexpression and a high degree correlation in terms of gene expression in DLBCL patients, suggesting that CYCLON and NPM1 could be co-regulated in DLBCL. However, although most of cases [71.2%] were double positive CYCLON(+) NPM1(+) in our cohort, we were not able to show strict association between CYCLON and NPM1 expression (p=0.35; Figure 2A). Moreover, CYCLON and NPM1 staining intensity, number of positive cells and subcellular localization were strictly independent. This discrepancy can be explained by the highly mobile and tunable nature of the proteins that can shuffle in and out of the nucleoli for CYCLON and between cytosol and nucleus for NPM1. CYCLON or NPM1 enrichment in a specific cellular compartment is most likely regulated by post-translational modifications and multiple interactions involving both protein and nucleic acids as previously reported for NPM1.
This suggests that despite the fact that CYCLON and NPM1 can physical interact in DLBCL cells, it seems that there is no strict co-regulation or functional association of the two proteins in DLBCL. This is further supported by the fact that NPM1 is a ubiquitous protein whereas CYCLON has a restricted tissue expression profile. However, we report here that concomitant expression of these particular CYCLON and NPM1 IHC staining patterns represent a very potent predictor of refractory DLBCL.
Some of these the elements have been added to the extended and remodeled version of the discussion section in the revised manuscript.
- There are several minor language issues, please check the manuscript carefully.
Manuscript has been checked carefully as recommended, we hope that it does not contain any more language issues.
- Please avoid the frequent use of superlatives.
We did our best to moderate the use of superlative in the manuscript.
- The introduction stated that the GCB subtype corresponds with an inferior survival while ABC subtype corresponds with inferior survival.
As mentioned in reviewer’s 1 reply, we sincerely apologize for that mistake, which has been corrected in the manuscript.
- The original referral articles referencing COO-classification are lacking
Original paper by Alizadeh et al, referencing COO-classification has been added to the manuscript’s bibliography section.
- DHX9 and HNRNPU at page 13 (sentence 401) were not mentioned anywhere else in the article.
We thank the reviewer for pointing at this inconsistency: a short sentence mentioning these other CYCLON interactants of interest was added to the result section.
- For readability, all figures should be greatly improved in editing
We have modified the way figures files were imported in the manuscript word document to improve overall resolution. This should greatly improve readability.
- Please add the numbers at risk to the KM-curves
This has been done.

Reviewer 3 Report
The manuscript reports the role of nuclear protein CYCLON in diffuse large B cell lymphoma (DLBCL). Using mass spec analysis, the authors identified a group of CYCLON binding proteins with a focus on NPM1, an abundant and ubiquitously expressed nucleolar phosphoprotein that functions as a molecular chaperone and shuttles between nucleus and cytoplasm. The authors performed immunohistochemistry evaluation of CYCLON and NPM1 and revealed that their co-expression is strongly associated with inferior prognosis in DLBCL. More specifically, alternative sub-cellular localizations of the proteins potentially predict treatment responses. In general, the study has the limited clinical samples and lacks in-depth mechanistic analyses. Many questions remain to address. For example, why is wild-type NPM1 relocalized in DLBCL cells? Is this sub-cellular localization a tumor specific compared to normal germinal center B cells? Is NPM1 expression required for cancer cell proliferation and survival? Is NPM1 required for CYCLON sub-cellular localization?
Author Response
Reply to reviewer 3
The manuscript reports the role of nuclear protein CYCLON in diffuse large B cell lymphoma (DLBCL). Using mass spec analysis, the authors identified a group of CYCLON binding proteins with a focus on NPM1, an abundant and ubiquitously expressed nucleolar phosphoprotein that functions as a molecular chaperone and shuttles between nucleus and cytoplasm. The authors performed immunohistochemistry evaluation of CYCLON and NPM1 and revealed that their co-expression is strongly associated with inferior prognosis in DLBCL. More specifically, alternative sub-cellular localizations of the proteins potentially predict treatment responses. In general, the study has the limited clinical samples and lacks in-depth mechanistic analyses.
Many questions remain to address.
- For example, why is wild-type NPM1 relocalized in DLBCL cells? Is this sub-cellular localization a tumor specific compared to normal germinal center B cells?
NPM1 cytosolic relocalization has only been reported as a consequence of expression of fusion proteins in ALCL/APL [1,2] or exon 12 frameshift mutations altering NPM1 nuclear localization (NLS) or export (NES) signals in AML [3]. Of note, no genetic defect of this type has ever been reported in DLBCL (or in any other hematological malignancy or solid tumor). Moreover, these mutations result in a 100% cytosolic NPM1 relocation (which has also been observed in some DLBCL cases, whereas no such mutations have ever been described in B-NHL), but does not result in the specific mixed pan-cellular NPM1 staining pattern associated with poor prognosis, which was never reported to date to our knowledge. We are currently investigating the molecular mechanisms that could explain these alternate localizations, but this functional characterization is beyond the scope of the study presented here, which aimed at studying the prognostic value of CYCLON and NPM1. At this stage, we can only postulate, that CYCLON and NPM1 alternate subcellular localizations in DLBCL can be due to alterations in protein-protein interactions (through cancer-related mislocalization of interacting partners), protein-targeting signals, transport machinery or post-translational modifications [4].
Of note, a nuclear NPM1 IHC staining was found in all control tissues used to set-up IHC staining procedure (normal lymph node, appendix, and testis), consistently with previously published and public IHC data from the Human Protein Tissue Atlas (https://www.proteinatlas.org/ENSG00000181163-NPM1/tissue).
Some of these elements have been integrated to the discussion section in the revised manuscript.
- Is NPM1 expression required for cancer cell proliferation and survival? Is NPM1 required for CYCLON sub-cellular localization?
NPM1 is a ubiquitous, oligomeric and multidomain protein that acts as a nucleolar organizer but also in multiple additional activities related to cell growth, homeostasis and stress response. NPM1 has a chaperone activity for ribosomal proteins and histones [5], as well as for transiently accumulated misfolded proteins in the nucleolus [6]. It has also been shown to regulate stability and localization of several tumor suppressor proteins including TP53 [7], thereby contributing to regulate apoptosis/cell proliferation balance. NPM1 has been further connected with the regulation of mitosis, replication, genome stability and transcription, which is consistent with the fact that NPM1 levels directly correlate with protein synthesis rate and proliferation status of the cell [5]. Accordingly, NPM1 is overexpressed in highly proliferative cells, and has been proved to be associated with several types of cancer, including here in DLBCL (Figure 1D).
Oncogenic fusion proteins involving NPM1 (NPM1–ALK, NPM–RARα or NPM1–MLF1) have also been described in ALCL [2] and in rare variants of AML [1]. Moreover, exon 12 NPM1 gene mutations leading to NPM1 protein relocation in the cytoplasm (NPM1c) are frequent found in 40% of AML [3,8]. NPM1-mutated AML have been recognized as a distinct entity in the last WHO myeloid neoplasm classification [1] related to favorable outcomes [8,9]. Interestingly, it has been show that NPM1c leukemia express distinctive stem cell–like gene expression pattern [10]. Moreover, NPM1 mutations were found in committed progenitors and differentiated myeloid cells in AML but were absent from the stem cell and lymphoid compartments [11]. This suggests that NPM1c induce aberrant progenitor self-renewal in myeloid progenitors that represents a critical step in the development of AML.
Overall, the role of NPM1 in tumorigenesis appears to be complex and probably context-specific, varying amongst different tumor types. Indeed, NPM1 is thought to exert both oncogenic and tumor suppressor roles, as it has been described to be overexpressed, mutated, aberrantly located or even deleted in tumors [12]. It is therefore very difficult to further speculate at this stage on the mechanisms driving and the consequences of NPM1 pan-cellular localization in DLBCL and their implications for treatment resistance.
As we initially identified NPM1 as a CYCLON interacting partner, we postulated that that NPM1 could be involved in regulating CYCLON alternate sub-nuclear localization. As a matter of fact, it has been recently described that NPM1 is involved in nucleolar structuration through multiple protein via R-tracts binding and rRNA interaction within liquid phase separation compartments [13]. CYCLON would represent a good candidate to participate to such functions as a disordered protein containing arginine-rich linear motifs. However, concomitant IHC analysis of both proteins did not allow to confirm this hypothesis. We can assume that that the regulatory hub controlling these relocalization events is not binary, but involves multiple other molecular actors that remain to be uncovered.
Some of these elements have been integrated to the discussion section in the revised manuscript.
References:
- Chen, Y.; Hu, J. Nucleophosmin1 (NPM1) abnormality in hematologic malignancies, and therapeutic targeting of mutant NPM1 in acute myeloid leukemia. Therapeutic advances in hematology 2020, 11, 2040620719899818, doi:10.1177/2040620719899818.
- Morris, S.W.; Kirstein, M.N.; Valentine, M.B.; Dittmer, K.G.; Shapiro, D.N.; Saltman, D.L.; Look, A.T. Fusion of a kinase gene, ALK, to a nucleolar protein gene, NPM, in non-Hodgkin's lymphoma. Science 1994, 263, 1281-1284, doi:10.1126/science.8122112.
- Falini, B.; Mecucci, C.; Tiacci, E.; Alcalay, M.; Rosati, R.; Pasqualucci, L.; La Starza, R.; Diverio, D.; Colombo, E.; Santucci, A.; et al. Cytoplasmic nucleophosmin in acute myelogenous leukemia with a normal karyotype. The New England journal of medicine 2005, 352, 254-266, doi:10.1056/NEJMoa041974.
- Wang, X.; Li, S. Protein mislocalization: mechanisms, functions and clinical applications in cancer. Biochimica et biophysica acta 2014, 1846, 13-25, doi:10.1016/j.bbcan.2014.03.006.
- Lopez, D.J.; Rodriguez, J.A.; Banuelos, S. Nucleophosmin, a multifunctional nucleolar organizer with a role in DNA repair. Biochimica et biophysica acta. Proteins and proteomics 2020, 1868, 140532, doi:10.1016/j.bbapap.2020.140532.
- Frottin, F.; Schueder, F.; Tiwary, S.; Gupta, R.; Korner, R.; Schlichthaerle, T.; Cox, J.; Jungmann, R.; Hartl, F.U.; Hipp, M.S. The nucleolus functions as a phase-separated protein quality control compartment. Science 2019, 365, 342-347, doi:10.1126/science.aaw9157.
- Colombo, E.; Marine, J.C.; Danovi, D.; Falini, B.; Pelicci, P.G. Nucleophosmin regulates the stability and transcriptional activity of p53. Nature cell biology 2002, 4, 529-533, doi:10.1038/ncb814.
- Verhaak, R.G.; Goudswaard, C.S.; van Putten, W.; Bijl, M.A.; Sanders, M.A.; Hugens, W.; Uitterlinden, A.G.; Erpelinck, C.A.; Delwel, R.; Lowenberg, B.; et al. Mutations in nucleophosmin (NPM1) in acute myeloid leukemia (AML): association with other gene abnormalities and previously established gene expression signatures and their favorable prognostic significance. Blood 2005, 106, 3747-3754, doi:10.1182/blood-2005-05-2168.
- Xu, L.H.; Fang, J.P.; Liu, Y.C.; Jones, A.I.; Chai, L. Nucleophosmin mutations confer an independent favorable prognostic impact in 869 pediatric patients with acute myeloid leukemia. Blood cancer journal 2020, 10, 1, doi:10.1038/s41408-019-0268-7.
- Vassiliou, G.S.; Cooper, J.L.; Rad, R.; Li, J.; Rice, S.; Uren, A.; Rad, L.; Ellis, P.; Andrews, R.; Banerjee, R.; et al. Mutant nucleophosmin and cooperating pathways drive leukemia initiation and progression in mice. Nature genetics 2011, 43, 470-475, doi:10.1038/ng.796.
- Jaiswal, S.; Fontanillas, P.; Flannick, J.; Manning, A.; Grauman, P.V.; Mar, B.G.; Lindsley, R.C.; Mermel, C.H.; Burtt, N.; Chavez, A.; et al. Age-related clonal hematopoiesis associated with adverse outcomes. The New England journal of medicine 2014, 371, 2488-2498, doi:10.1056/NEJMoa1408617.
- Karimi Dermani, F.; Gholamzadeh Khoei, S.; Afshar, S.; Amini, R. The potential role of nucleophosmin (NPM1) in the development of cancer. Journal of cellular physiology 2021, doi:10.1002/jcp.30406.
- Mitrea, D.M.; Cika, J.A.; Guy, C.S.; Ban, D.; Banerjee, P.R.; Stanley, C.B.; Nourse, A.; Deniz, A.A.; Kriwacki, R.W. Nucleophosmin integrates within the nucleolus via multi-modal interactions with proteins displaying R-rich linear motifs and rRNA. eLife 2016, 5, doi:10.7554/eLife.13571.